# Japanese Encephalitis Virus Genotype III Strains Detection and Genome Sequencing from Indian Pig and Mosquito Vector

**DOI:** 10.3390/vaccines11010150

**Published:** 2023-01-10

**Authors:** Seema R. Pegu, Pranab Jyoti Das, Joyshikh Sonowal, Gyanendra Singh Sengar, Rajib Deb, Ajay Kumar Yadav, Swaraj Rajkhowa, Manjisa Choudhury, Baldev R. Gulati, Vivek Kumar Gupta

**Affiliations:** 1ICAR-National Research Centre on Pig, Rani, Guwahati 781131, India; 2ICAR-Indian Veterinary Research Institute, Izatnagar, Bareilly 243122, India; 3ICAR-National Research Centre on Equines, Hisar 125001, India

**Keywords:** Japanese encephalitis virus, pig, mosquito, genome sequencing, phylogeny, polymorphism

## Abstract

Japanese encephalitis viruses (JEVs) are globally prevalent as deadly pathogens in humans and animals, including pig, horse and cattle. Japanese encephalitis (JE) still remains an important cause of epidemic encephalitis worldwide and exists in a zoonotic transmission cycle. Assam is one of the highly endemic states for JE in India. In the present study, to understand the epidemiological status of JE circulating in pigs and mosquito, particularly in Assam, India, molecular detection of JEV and the genome sequencing of JEV isolates from pigs and mosquitoes was conducted. The genome analysis of two JEV isolates from pigs and mosquitoes revealed 7 and 20 numbers of unique points of polymorphism of nucleotide during alignment of the sequences with other available sequences, respectively. Phylogenetic analysis revealed that the isolates of the present investigation belong to genotype III and are closely related with the strains of neighboring country China. This study highlights the transboundary nature of the JEV genotype III circulation, which maintained the same genotype through mosquito-swine transmission cycles.

## 1. Introduction

Japanese encephalitis (JE) is a common mosquito born flaviviral encephalitis in Southeast Asian countries. JE virus (JEV) belongs to the family *Flaviviridae* and is principally a disease occurring in rural agricultural areas maintained in an enzootic cycle involving water birds, domestic pigs and zoophilic *Culex* mosquitoes [1]. Humans and horses are the dead-end hosts, in which JEV causes acute encephalitis with the onset of acute febrile illness and alterations in mental status such as confusion, inability to talk, disorientation, seizures and coma [2]. Hence, JE is a cause of public health concern [2,3]. Pigs and water birds are the amplifying hosts of the JEV without any clinical signs, except abortion and still births in pregnant sows and gilts [3]. In 1935, JEV was isolated from Japan [4] and in 1955, it was detected in infected patients admitted to the Christian Medical College and Hospital in Vellore, Tamil Nadu, India [5]. In the subsequent years, the virus has been found to be active in almost every part of India, and outbreaks have been reported regularly from Andhra Pradesh, Assam, Bihar, Goa, Haryana, Karnataka, Kerala, Maharashtra, Manipur, Orissa, Pondicherry, Tamil Nadu, Uttar Pradesh and West Bengal [6]. The incidence of Japanese encephalitis in recent times has shown an increasing trend in India and has become a major public health problem [2,7]. Assam is the north-eastern (NE) state of India and has the largest pig population in the country with abundance of rainfall and rice cultivated land, making Assam the most vulnerable state for JE incidences [2,8,9]. In Assam, JEV was first reported in the year 1976 from Lakhimpur district and since then incidences of JE have markedly increased [10,11]. Furthermore, the presence of JEV infected pigs increases the chances of spillover of infection to humans, especially when the mosquito density is high. For human populations in JE endemic regions in India, immunization with the SA-14-14-2 vaccine began in 2006 [12,13]. However, there are no specific vaccines available for animals in India. The SA-14-14-2 vaccine was imported from China until 2013 and after that, indigenously developed inactivated Vero cell-derived vaccine JENVAC has been introduced in India with effective results [6].

Previous studies indicate that JEV can be classified into five genotypes (I to V) based on the diversity of the nucleotide sequence of E protein gene, with most isolates classified as genotype GI or genotype GIII [14]. Although numerous encephalitis aetiology studies have confirmed the importance of this virus, little information on the genotype circulating strain has been reported from this part of India in the amplifying host of JE. The resurgence of JE cases in the human population of India over the last few years highlights the need for JEV surveillance in the amplifying host. In this study, we report the detection and next generation genome sequencing of two strains of JE virus from the pig and mosquito vector and make a comparison with recent isolates of JE virus from pigs and mosquitoes.

## 2. Materials and Methods

### 2.1. Collection of Samples

The selection of the study area and collection of pig samples were based on a pilot survey on the availability of pig populations, pig farms and slaughter points in Assam’s Kamrup and Jorhat districts [15]. In Figure 1, the area of study is depicted. A total of 112 nos. of pig samples (tissue, cerebrospinal fluid and blood) were collected from 8 pig farms and 4 slaughter points in specified districts of Assam from 2018 to 2020. A suitable questionnaire was prepared for collecting baseline information from the pig farms/slaughter points regarding age, breed, parity, no. of piglet born, any abortion or stillbirth cases, any other disease condition in the farm, etc.

Simultaneously, a total of 8124 mosquitoes was collected from Kamrup and Jorhat districts in accordance with the areas of pig farms during over dusk (5.00 PM to 7.00 PM). Mosquitoes were collected by aspirator, sweeping net and with the help of a CDC trap light inside and outside the pig farm. The collected female mosquitoes were sorted, preserved in separate plastic vials and identified under stereoscopic binocular microscope following taxonomic key.

### 2.2. RNA Extraction, Synthesis of cDNA and Amplicon Generation

Viral RNA of JEV was extracted from blood and tissue samples collected from pig and mosquito samples (mosquitoes were pooled at approx. 30 mosquitoes/pool) using QIAamp Viral RNA Mini Kit (QIAGEN, Hilden, Germany) as per manufacturer’s instruction. RNA was converted into complementary DNA (cDNA) using a RevertAid First Strand cDNA Synthesis Kit (Cat no: K1622, Thermo Scientific™ (Waltham, MA, USA), Lithuania-European Union) with random hexamer primers. The synthesized cDNA was used to amplify the partial envelope protein gene of JEV @ 236 bp by PCR using the forward primer 5′-TTACTCAGCGCAAGTAGGAGCGTCTCAAG-3′ and reverse primer 5′-ATGCCGTGCTTGAGGGGGACG-3′ [16]. A specific real-time RT-PCR assay was used for the detection of JEV gene. Cycling conditions comprised initial denaturation 95 °C for 10 s, 40 cycles at 95 °C for 15 s and 55 °C for 30 s and amplification curves were visualised in AriaMx v1.7 software. All the samples detected by real-time PCR were subjected to conventional PCR, as described previously. Conventional PCR amplification was performed in thermal cycler under the following conditions: cycle at 95 °C for 2 min (initial denaturation), 40 cycles (denaturation at 95 °C for 30 s, annealing at 65 °C for 30 s, and extension at 72 °C for 35 s), and cycle of final extension at 72 °C for 10 min. Five microliters of amplified PCR products were separated by 2% ethidium bromide strained agarose gel electrophoresis at 100 V for 25 min. An amount of 100 bp plus DNA Marker (Cat no.: SM1153, Thermo Scientific) was used as standard and the amplified products were visualized using an ultraviolet light transilluminator. Simultaneously, genotypic selection of positive JEV samples were confirmed by genotype specific reference PCR primers, as per Yang et al. [17].

### 2.3. Next-Generation Sequencing (NGS)/Genome Sequencing

The genome sequencing was done from two positive samples from pig and one positive sample from mosquito pool. Genome sequencing of the positive viral RNA samples were outsourced (Genotypic Technology Pvt. Ltd., India). Library preparation and sequencing were implemented with Illumina-compatible NEBNext^®^ Ultra™ II using Directional RNA Library Prep Kit (New England BioLabs, MA, USA). Viral total RNA was taken for fragmentation and priming. Fragmented and primed RNA was further subjected to first-strand synthesis, followed by second-strand synthesis. The double-stranded cDNA was purified using JetSeq Beads (Bioline, Cat # BIO-68031). Purified cDNA was end-repaired, adenylated and ligated to Illumina multiplex barcode adapters (Universal adapter: 5′-AATGATACGGCGACCACCGAGATCTACACTCTTTCCCTACACGACGCTCTTCCGATCT-3′ and Index adapter: 5′-GATCGGAAGAGCACACGTCTGAACTCCAGTCAC [INDEX] ATCTCGTATGCCGTCTTCTGCTTG-3′) as per NEBNext^®^ Ultra™ II Directional RNA Library Prep protocol, followed by second-strand excision using USER enzyme at 37 °C for 15 min. Adapter ligated cDNA was purified using JetSeq Beads and was subjected to 11 cycles for indexing (98 °C for 30 s, cycling (98 °C for 10 s, 65 °C for 75 s) and 65 °C for 5 min) to enrich the adapter-ligated fragments. The final PCR product (sequencing libraries) was purified with JetSeq Beads, followed by library quality control check. Illumina-compatible sequencing libraries were quantified by Qubit Fluorometer (Thermo Fisher Scientific) and the fragment size distribution was analyzed on Agilent 2200 Tape Station.

### 2.4. Genomic Alignment, Single Nucleotide Polymorphism and Phylogenetic Analysis

JEV genome sequences were obtained from the NCBI database and the accession no. of all the sequences are presented in Appendix A. The JEV detected from mosquito and pig were analyzed independently in two groups. Our samples were compared to other polyprotein sequences of JEV strains in the database using the ClustalW multiple alignment tool in MegAlign software. The aligned sequences underwent SNP identification and genetic relationship determination with the JEV strains of mosquito and pig of present study, separately. We also determined the common nucleotide mutation positions present in the JEV isolate from pig of India, separately. To establish the genetic relationship of JEV isolates from pig and mosquito, we constructed phylogenetic tree based on whole polyprotein gene sequences.

## 3. Results

### 3.1. Laboratory Testing of JE Case Specimens

#### 3.1.1. Gross and Histopathological Findings

Our study investigated and found that a sow in her first farrowing gave birth to four stillborn piglets at full term (Figure 2A) and tested positive for JEV. The vector mosquito collected from the pig sheds and nearby drainage and waterlogged areas revealed the prevalence of *Culex tritaeniorhynchus* mosquitoes in abundance during the evening and early morning periods (Figure 2B). The post-mortem examination of all the stillborn piglets showed varying degrees of congestion in the lung, liver, heart, spleen and lymph nodes (Figure 2C). Upon opening the head, two piglets showed encephalitis in the cerebrum (Figure 2D). In a histopathological investigation, the most significant microscopic lesions found in the brain were neuronophagi, satellitosis and vacuolation in the cerebrum (Figure 2E), as well as lymphoid depletion in the peripheral lymph nodes (Figure 2F).

#### 3.1.2. Molecular Diagnosis/Viral Gene Detection (RT-PCR and Real Time RT-PCR)

Out of 112 pig samples, JEV was positive in 11 samples (Table 1). A total of 271 pools of mosquitoes were constituted by the end of collection based of species and sex and were tested for JEV by real-time RT-PCR. Out of 271 pools, two pools of *Culex tritaeniorhynchus* and *Culex gelidus* from female mosquito pools were found to be positive (Table 2). For the Ct value cut-off, the cloned JEV plasmid product was serially diluted 10-fold from 2.89 × 10^8^ copies/μL and determined the detection limits of the qPCR. We have used 10^−7^ dilution of a representative positive sample of JEV whose Ct value is ~32.17 and represent the 2.89 × 10^1^ copies/μL. Beyond this dilution, we found Ct value close to 38, which is negligible or could be considered as negative. Therefore, we selected 2.89 × 10^1^ copies/μL (32.17) as a cutoff point for JEV detection in real-time qPCR. The representative screened samples were shown in Figure 3 (Real time PCR) and Figure 4 (Conventional PCR). The genotype-specific PCR revealed all the positive samples belonged to genotype III (Appendix A).

### 3.2. Genome Sequencing

The genome of the JEV detected in a pig sample and in mosquito pool sample from Assam (JEV/Assam/Pig), India was sequenced. The raw sequence data of the viral genome isolated from pig tissue sample was assembled and annotated using bioinformatics software and was found to be 10,966 nucleotides(nt) long with 97 nt at the 5′ untranslated region (UTR), as well as a 10,299-nt open reading frame (single) corresponding to 3,432 amino acids excluding stop codon and 570 nt of the 3′ UTR.

The positive mosquito pool samples were also sequenced and analyzed to be found to be 10,965 nucleotides(nt) long with 96 nt at the 5′ untranslated region (UTR), as well as a 10,299-nt open reading frame (single) corresponding to 3,432 amino acids excluding stop codon and 570 nt of the 3′ UTR.

### 3.3. Phylogenetic Analysis

The sequence alignments and phylogenetic analysis revealed that the virus belongs to genotype III. The NCBI BLAST analysis revealed that the JEV/Assam/Pig genome sequence shared highest similarity (99.2%)with Indian JEV strain isolated from pig (MT232844) followed by 98.3% of China variant (MN544780) (Table 2). The genome of the mosquito vector showed a high nucleotide homology (98%) to reference strain 057434 (EF623988). The NCBI Blast mosquito sequences revealed 98.5% (MH385014:MT254426) of China. Phylogenetic analysis of both amplifying host and vector showed region specific similarity with the host (Figure 5A) and vector (Figure 5B). Both the sequences of JEV identified in this study also showed the same trend of having the greatest similarity in mosquito sequences (ON875960 and MZ702743) (Figure 5C). The radial phylogenetic analysis of all the sequences shows that in the distance matrix of all vectors and hosts under this study, Assam mosquitoes shows the highest similarity of 99.4% with China pigs, and mosquitoes and Assam pigs showed the highest similarity with Indian pigs and Assam mosquitoes (Appendix A).

### 3.4. Unique Polymorphism of Nucleotides

A total of 20 number unique mutations of nucleotides were found in polyprotein gene of JEV isolated from mosquito, ON875960 (Figure 6A, Appendix A). In case of JEV isolated from pigs, 7 numbers of unique mutation of nucleotides were found in polyprotein gene of MZ702743.1 (Figure 6B, Appendix A). Additionally, 15 numbers of unique polymorphism of nucleotides were commonly observed in polyprotein gene of JEV strains isolated from pigs of India (MZ702743.1 and MT232844.1) from other reference genes (Figure 6C, Appendix A). The mutations result in unique substitutions of 4 numbers amino acids in MZ702743.1 (Appendix A) and 8 numbers of amino acids in ON875960 (Appendix A).

## 4. Discussion

JEV is an RNA virus that has a high potential for evolution due to its lack of repair mechanisms that would otherwise act during the replication of its genome [18,19]. Extensive surveillance of amplifying hosts and vectors assists in understanding geographical migration and genotype shift in JEV [20]. The current work represents a constrained example of JEV molecular evolution in nature. JEV is still a serious but neglected public health problem in India and is the most frequent aetiology of meningeo-encephalitis in humans of several Asian countries [21,22]. In the north-eastern region of India, large JE epidemics occur in humans during the monsoon season, when mosquito density is abundant. There has been a recent increase of JE/Acute Encephalitis Syndrome (AES) cases in Assam, which have migrated from upper Assam to every adjoining district; there have been reports of confirmed cases and mortality throughout the state [23]. In the present study, from 2018 to 2020, in a total of 112 pig samples collected, we detected JEV by RT PCR in 11 pig samples. A total of 8124 mosquitoes were collected from the Jorhat and Kamrup districts in accordance with the areas pig samples were collected. A total nine mosquito species were identified, including *Culex tritaeniorhynchus*, *Culex vishnoi*, *Culex pseudovishnoi*, *Culex gelidus*, *Culex whitemoorei*, *Mansonia.* The most prevalent species was *Culex tritaeniorhynchus*, followed by *Culex gelidas* and the *Culex vishnui*, *Culex pseudovishnoi* and *Culex quinquefasciatus* subgroup. A total of 278 pools of mosquitoes consisting of 20–30 mosquitoes in one pool were constituted every fortnight of collection based on species and tested for JEV by RT-PCR. Out of all the tested samples, only one pool of *Culex tritaeniorhynchus* captured in August 2019 from Kamrup rural was found to be positive subjected for WGS.

The polyprotein gene is the complete ORF of JEV flanked by 5′ and 3′ untranslated regions (UTRs) and the polyprotein consisting of structural proteins such as capsid [C], membrane [prM/M], envelope [E] and non-structural proteins such as NS1, NS2A, NS2B, NS3, NS4A, NS4B and NS5 [24]. The polyprotein gene sequences of JEV stain isolated from pig and isolated from mosquito were compared separately for the genetic relationship via phylogenetic tree with other stains of respective JEV isolated from pig and mosquitoes, and showed that a number of nucleotide substitutions (unique polymorphism) were scattered throughout the polyprotein gene. The analysis of JEV from pig revealed the circulation of JEV genotype III in the pig population of Assam, India; this is region specific. This is India’s first study to disclose the entire genomic sequences of JEV isolated from mosquitoes. The majority of JEV isolates in India from different sources are genotype I or genotype III [25]. Genotype III is also widespread in Asian countries such as China, Japan, Nepal, Philippines, South Korea, Sri Lanka, Taiwan and Vietnam [26]. However, the differences in the sequence variation did not result in changes in the genotype among the strains, but virulence might be varying. Thus the polyprotein gene sequence is suitable for genotyping as well as monitoring gene evolution. Despite this, the E gene sequences of JEV are reliable phylogenetic markers that may be used to determine all five genotypes (Genotypes I to V) [17,27]. The JEV isolates from pigs and mosquitoes of the present study showed less mutation and variation in the point of origin analysis of different JEV sequences, indicating that our isolates were the most ancestral. The phylogenetic analysis of JEV (MZ702743.1) isolated from pigs showed that they were closely related to the isolates of India (MT232844.1) followed by the isolate of neighboring country China (MN544780.1). On the other hand, analysis indicated that JEV isolated from mosquito for the first time from India (ON875960.1) was closely related to the strains of neighboring country China (MH385014.1 and MT254426.1). This investigation indicates the transboundary nature of the JE. Since north-eastern states of India are located on the border of China, Myanmar, Bhutan and Bangladesh, this investigation not only provides a basis for the prevalence of JEV in this region, but also for neighboring countries. Thus the data from this study may aid in the design of effective control strategies for each of these regions as well as provide future impact on development of new therapeutics and vaccine against JEV, including genotype III strains.

## 5. Conclusions

In conclusion, our report clearly shows that JEV genome sequencing from the pig and mosquito vectors belongs to Genotype III circulating in this region, which is same with the virus circulating in humans. It is worth mentioning that *Culex tritaeniorhynchus* is the most important vector for JEV transmission in India. Evidence of JEV transmission in this region (in pigs and mosquitoes) may have potential implications for future spread of JEV in human and other animal populations. Additional survey of JEV in pig, human and mosquito vector in NE region India is necessary to better understand strategic control of the disease.

## Figures and Tables

**Figure 1 vaccines-11-00150-f001:**
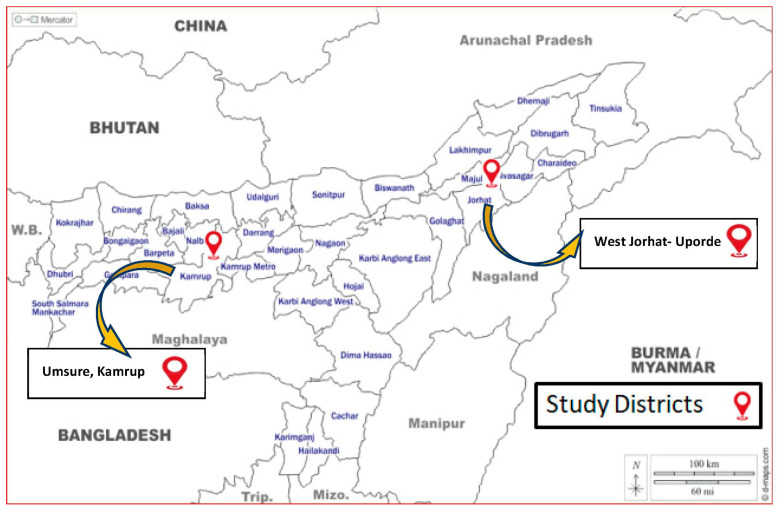
Demonstration of the geographical area where samples or specimens were collected for the current study. Here, the Assam, India, districts of Kamrup and Jorhat were chosen due to their high sales of pigs and pig products and as they are JE hotspot locations.

**Figure 2 vaccines-11-00150-f002:**
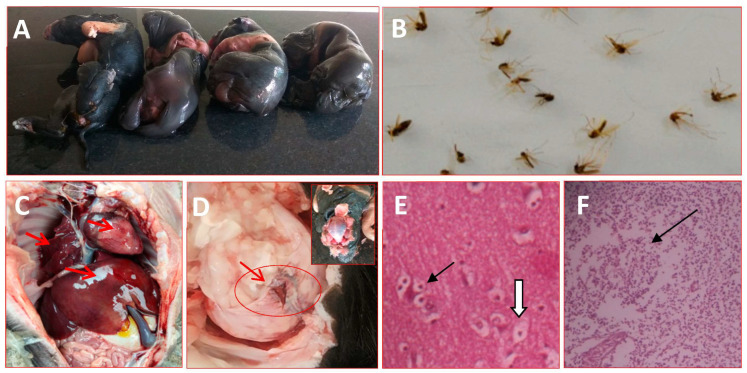
Japanese encephalitis (JE) case specimens of the current study. Here, (**A**) Stillborn fetuses in a sow due to JE infection during pregnancy, (**B**) A pool of *Culex tritaeniorhynchus* mosquito, a major vector of JEV transmission, (**C**) Congestion in the internal organs (arrow), (**D**) Encephalitis in the brain in a stillborn foetus affected with JE (arrow), (**E**) Neuropathology in the cerebrum of a JEV-infected stillborn piglet. Neuronophagia and satellites (arrow), neuronal vacuolation (arrowheads) hematoxylin and eosin stained, 20× magnification. (**F**) Lymphoid depletion in the lymphoid follicle of lymph node (arrow) hematoxylin and eosin stained, 20× magnification.

**Figure 3 vaccines-11-00150-f003:**
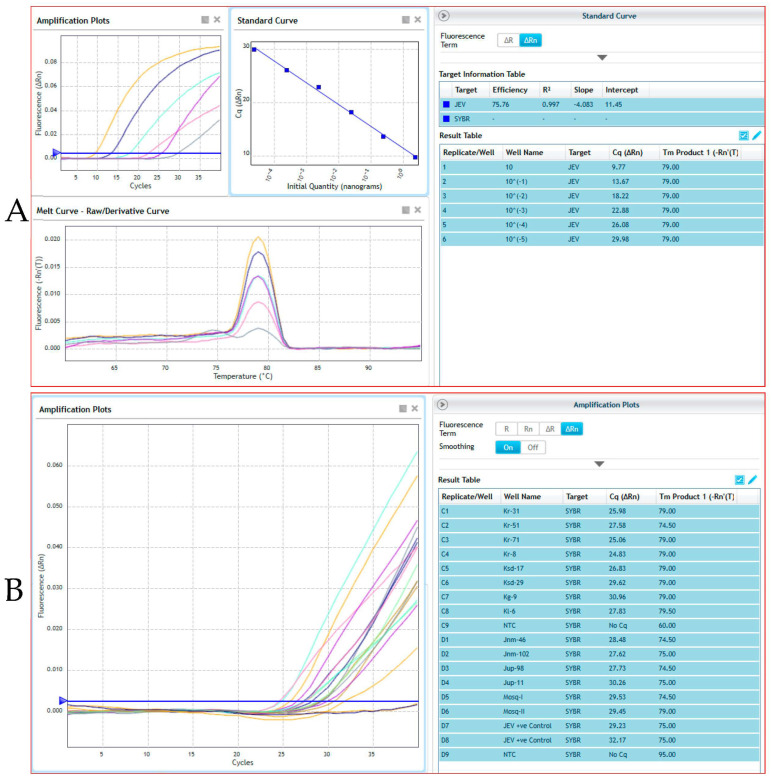
Real time PCR for detection of Japanese encephalitis virus (JEV) from different pig and mosquito samples. Here, (**A**) shows the standard curve analysis of a representative JEV positive control sample, where 10-fold dilution factor of the sample represents 75.76% of efficiency upto 6th dilution (10^−5^); (**B**) shows the presence of JEV in representative test pig and mosquito samples. These indicated that the positive samples were contains JEV cDNA/RNA in each respective reaction. The 6th (well-D7) and 7th (well-D8) dilution of samples were used as representative positive control for detection of JEV concentration in the present study.

**Figure 4 vaccines-11-00150-f004:**
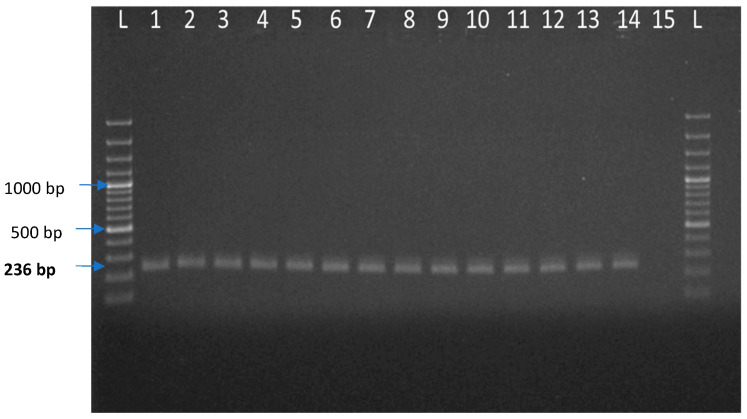
Agarose gel electrophoresis of JEV partial E gene amplified products isolated from pig and mosquito. Here, L: 100 bp plus ladder, 1: Positive control of JEV, 2 to 12: Test samples of JEV isolated from pig sources, 13 and 14: Test samples of JEV isolated from mosquito, 15: Negative control.

**Figure 5 vaccines-11-00150-f005:**
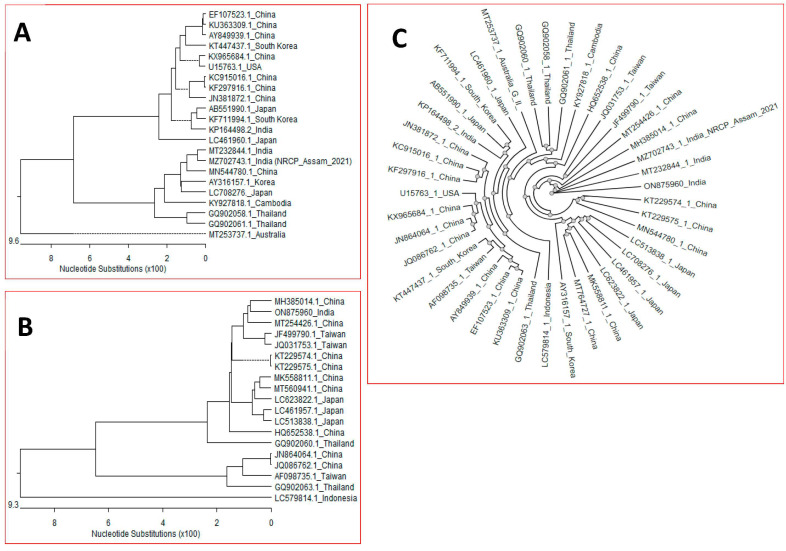
Representation of phylogenetic profile of the JEV isolates, (**A**) isolates from pigs, and (**B**) isolates from the vector *Culex tritaeniorhynchus* mosquito based on a comparison of the whole polyprotein gene, (**C**) Phylogenetic tree of JEV isolates from pig & mosquito, point of origin.

**Figure 6 vaccines-11-00150-f006:**
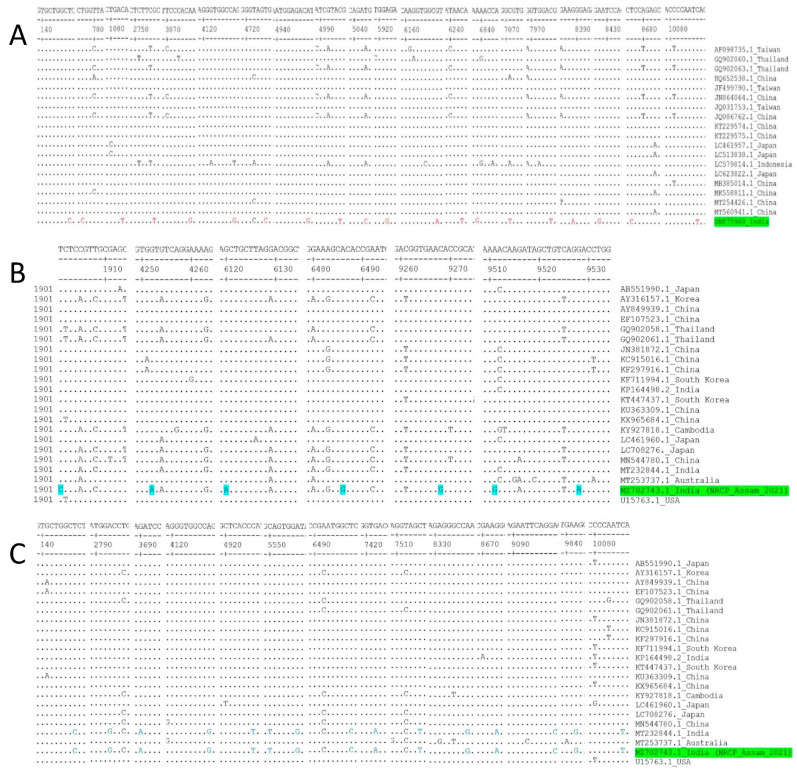
Nucleotide sequence alignment of the polyprotein genes of JEV isolates from mosquito (**A**), pig (**B**) and unique nucleotide mutations identification in Indian isolate of current study. (**C**) Common and unique nucleotides mutation of polyprotein genes of JEV isolates from two Indian strain. Residues identical to the consensus are indicated by dots.

**Table 1 vaccines-11-00150-t001:** Description of JEV positive samples collected from pigs from Jorhat and Kamrup District.

Sample Details					Results		Genotype
Animal ID	Age	Sex	Type	Location	Real-Time RT-PCR	Conventional RT-PCR	
Kr-31	62d	F	Blood	Rani, Kamrup	+ve	+ve	GIII
Kr-51	75	F	Blood	Rani, Kamrup	+ve	+ve	GIII
Kr-71	114	F	Blood	Rani, Kamrup	+ve	+ve	GIII
Kr-8	Foetus	F	Tissue	Rani, Kamrup	+ve	+ve	GIII
Ksd-17	Foetus	F	Tissue	Sundubi, Kamrup	+ve	+ve	GIII
Ksd-29	Foetus	F	Tissue	Sundubi, Kamrup	+ve	+ve	GIII
Kg-9	8M	M	Tissue	Gorchuk, Kamrup	+ve	+ve	GIII
Kl-6	1YR	M	Tissue	Lakhra, Kamrup	+ve	+ve	GIII
Jnm-46	9m	m	Tissue	Namdeuri, West Jorhat	+ve	+ve	GIII
Jnm-102	12m	F	Tissue	Namduri, West Jorhat	+ve	+ve	GIII
Jup-98	8m	F	Tissue	Upper deuri, West Jorhat	+ve	+ve	GIII
Jup-11	8m	F	Tissue	Upper deuri, West Jorhat	+ve	+ve	GIII

**Table 2 vaccines-11-00150-t002:** List of mosquito species identified with result of JEV screening.

Mosquito Species	Sex	Total Sample	Pools	Location	RT-PCR	PCR	Genotype
*Culex tritaeniorhynchus*	F	4236	141	Jorhat & Kamrup	Positive in one pool	Positive in one pool	GIII
*Culex gelidus*	F	1563	52		Positive	Positive	GIII
*Culex quinquefasciatus*	F	121	6		negative	negative	-
*Culex vishnoi*	F	1192	40		negative	negative	-
*Culex pseudovishnoi*	F	714	23		negative	negative	-
*Culex whitemorei*	F	75	4		negative	negative	-
*Mansonia*	F	98	5		negative	negative	-
*Armegeres*	F	73	4		negative	negative	-
*Anopheles*	F	52	3		negative	negative	-

## Data Availability

The data that support this study will be shared upon reasonable request to the corresponding author.

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
