# Peer review of "Japanese Encephalitis Virus Genotype III Strains Detection and Genome Sequencing from Indian Pig and Mosquito Vector"

_vaccines, 2023, doi:10.3390/vaccines11010150_

Round 1

Reviewer 1 Report

The manuscript by Pegu et al describes the detection and WGS of JEV from pigs and mosquitoes in Northeastern India. The manuscript describes the methods for surveillance and sequencing well, citing the appropriate literature. Overall the manuscript covers an essential topic of JEV surveillance and its transboundary implications.

Although the study is conducted well, there are minor comments-

- Culex needs to be italicized at multiple places

- Legends for figures seem to be missing the last sentence(s)

- Figures 2 & 5 have parts of the 'A', 'B' etc missing.

Author Response

Response to Reviewer’s comments

Reviewer 1:

The authors would like to express their sincere thanks to the Editor and the anonymous reviewers for their great comments and suggestions. We are also sincerely thankful to Editor for providing an opportunity to revise and resubmit this manuscript for publication in Vaccines.

The manuscript by Pegu et al describes the detection and WGS of JEV from pigs and mosquitoes in Northeastern India. The manuscript describes the methods for surveillance and sequencing well, citing the appropriate literature. Overall the manuscript covers an essential topic of JEV surveillance and its transboundary implications.

Although the study is conducted well, there are minor comments-

- Culex needs to be italicized at multiple places.

Response: Corrections made in the manuscript as desired by the reviewer

- Legends for figures seem to be missing the last sentence(s).

Response: Corrections made in the manuscript as desired by the reviewer

- Figures 2 & 5 have parts of the 'A', 'B' etc missing.

Response: Corrections made in the manuscript as desired by the reviewer

Reviewer 2 Report

Introduction

·         The introduction as written provides little to know information on the incidence of JEV in India or its neighboring countries. It is not enough to say incidence has increased over the years, authors need to provide data to support that claim and justify the need for the study

·         Authors need to provide information on the circulating genotypes in India and/or its environs. It is not enough to say there is little information on the genotypes, what information is classified as little?

Method

·         Paragraph under collection of samples (2.1) needs to be re-written for clarity. Authors should provide a reference for the pilot survey they are talking about.

·          Section 2.2: How did authors specifically isolate only virus RNA: did authors extract total RNA?

·         What was the cut off point for JEV detection in the real time PCR?

·         How’s the phylogenetic tree different for the NJ written In the paragraph?

Results

·         Are authors inferring that the stillbirths were as a result of JEV infection? If so were the stillborn screened for JEV? What was the result?

·         Was the screening of the sow done before during or after the birth? If it was after, how long after was the screening done?

·         Was the 11 positive pigs (and 4 positive mosquito pools) positive by qPCR, RT-PCR or both

·         Were all pigs and mosquitoes collected from the same shed/farm?

·         Why did authors sequence only 1 of the 11 positive pig samples? Is that representative of the JEV circulating in the city?: No.

·         How did authors confirm the genotype of all the detected JEV if only 1 out of 11 was sequenced?

·         Did authors perform RACE to determine the terminal sequences of JEV detected?; If not authors can’t assume they have sequenced the complete/entire Genome of the detected JEV.

Author Response

Response to Reviewers comments

Reviewer 2:

The authors would like to express their sincere thanks to the Editor and the anonymous reviewers for their great comments and suggestions. We are also sincerely thankful to Editor for providing an opportunity to revise and resubmit this manuscript for publication in Vaccines.

Introduction

  • The introduction as written provides little to know information on the incidence of JEV in India or its neighboring countries. It is not enough to say incidence has increased over the years, authors need to provide data to support that claim and justify the need for the study

Response: Authors added more information about JEV epidemiology in the manuscript as suggested by the reviewer.

  • Authors need to provide information on the circulating genotypes in India and/or its environs. It is not enough to say there is little information on the genotypes, what information is classified as little?

Response: Authors added information of JEV genotype I to III circulating in India and neighboring centuries.

Method

  • Paragraph under collection of samples (2.1) needs to be re-written for clarity. Authors should provide a reference for the pilot survey they are talking about.

Response: The sentence has been rephrased, and the reference to the previous study on the prevalence of Japanese encephalitis in four districts of Assam has been added.

  • Section 2.2: How did authors specifically isolate only virus RNA: did authors extract total RNA?

Response: We have used QIAamp Viral RNA Mini Kit (QIAGEN, Hilden, Germany) as per manufacturer’s instruction.

  • What was the cut off point for JEV detection in the real time PCR?

Response: We have used 10-7 dilution of a representative positive sample of JEV which ct value is ~32.17.

The cloned JEV plasmid product was serially diluted 10-fold from 2.89 × 108 copies/μL to 2.89 × 10copies/μL to determine the detection limits of the qPCR. We have used 10-7 dilution of a representative positive sample of JEV whose Ct value is ~32.17 and represent the 2.89 × 10copies/μL. Beyond this dilution we found Ct value close to 38 which is negligible or could be considered as negative. Therefore, we selected 2.89 × 10copies/μL (32.17) as a cut off point for JEV detection in real time qPCR.

  • How’s the phylogenetic tree different for the NJ written In the paragraph?

Response:  Neighbor joining is a bottom-up (agglomerative) clustering method for the creation of phylogenetic trees. Corrections made in the manuscript as desired by the reviewer.

Results

  • Are authors inferring that the stillbirths were as a result of JEV infection? If so were the stillborn screened for JEV? What was the result?

Response: The severe pathological lesions observed in the brain of the stillborn foetuses were caused by JEV associated reproductive failure. In our study investigated and found that a sow in her first farrowing gave birth to four stillborn piglets at full term and the sow and the stillborn piglets were tested positive for JEV.

  • Was the screening of the sow done before during or after the birth? If it was after, how long after was the screening done?

Response: The sow was screened after birth of stillborn foetuses. The animal was rescreened after 15th day of first screen but found negative for JEV.

  • Was the 11 positive pigs (and 4 positive mosquito pools) positive by qPCR, RT-PCR or both

Response: Yes, according to RT-PCR and conventional PCR, 11 pig samples and 2 pooled mosquito samples were positive.

  • Were all pigs and mosquitoes collected from the same shed/farm?

Response: Mosquitoes were collected from Kamrup and Jorhat districts in accordance with the areas of pig farms during the dusk period of the day.

  • Why did authors sequence only 1 of the 11 positive pig samples? Is that representative of the JEV circulating in the city?

Response:  A representative sample of JEV was sequenced. The sequencing of too many samples was costly for the project (not under the project budget).

  • How did authors confirm the genotype of all the detected JEV if only 1 out of 11 was sequenced?

Response: The Whole genome sequenced JEV isolates (isolated from pigs and mosquito) were confirmed by NGS analysis. Although, we have confirmed the 13 (11+2) positive samples with genotype III specific PCR. The genotype III specific reference PCR primers were given below (the data not shown in manuscript, a separate supplementary file is added along with the manuscript).

(Reference primer of http://dx.doi.org/10.4167/jbv.2016.46.4.231)

Forward: CCGATCGTCTCCGTTGCGAGCC

Reverse: TTTGGCCTTCTTAGCCACAG

Product size: 451 bp

  • Did authors perform RACE to determine the terminal sequences of JEV detected?; If not authors can’t assume they have sequenced the complete/entire Genome of the detected JEV.

Response: Whole genome sequencing was approached with illumina NGS and WGS sequence analysis steps. The samples were outsourced for WSG with NGS approached.

Reviewer 3 Report

Japanese encephalitis virus (JEV) has been recognized as a major pathogen causing viral encephalitis in Asian countries. Since 1990’s, genotype replacement (III to I) has been observed in many countries. To develop better strategies to control JEV infection, understanding current epidemiological situation in pigs, mosquitoes as well as humans. Authors detected full viral genomes from pig and mosquito. Several points were raised.

1.      Is it possible to provide more detailed JEV epidemiology in India (number of cases) ?

2.      Please mention licensed vaccines in India.

3.      Is JEV vaccines for veterinary use available in India? If yes, please mention its coverage or something like that.

4.      Materials and Table 1: Which tissues of pigs did authors obtain? Can authors clarify?

5.      Table 1: How did authors determine the genotype? Did authors check sequence of E genes of all positive samples? Please clarify.

6.      Fig. 5: The figure legend is missing.

7.      Fig. 5: If authors can provide isolated year and its source (mosquito, human…) for each data in the phylogenetic tree, that would be quite useful.

8.      Fig. 6: Were these substitutions synonymous or nonsynonymous ?

9.      Fig. 6: If authors can provide the viral proteins containing these substitutions in the figure , it should be helpful.

10.   Table 2: If I understand correctly, conventional PCR was conducted only for real-time PCR positive sample. However, table 2 contained the sample for real-time negative and conventional PCR positive.

11.   I cannot find supplementary data.

12.   Discussion: Please provide discuss on the genotypic epidemiology in India. Several papers describing that genotype III and I are found, but genotype III is recognized as dominant, can be found.

Author Response

Response to Reviewers comments

Reviewer 3:

The authors would like to express their sincere thanks to the Editor and the anonymous reviewers for their great comments and suggestions. We are also sincerely thankful to Editor for providing an opportunity to revise and resubmit this manuscript for publication in Vaccines.

Japanese encephalitis virus (JEV) has been recognized as a major pathogen causing viral encephalitis in Asian countries. Since 1990’s, genotype replacement (III to I) has been observed in many countries. To develop better strategies to control JEV infection, understanding current epidemiological situation in pigs, mosquitoes as well as humans. Authors detected full viral genomes from pig and mosquito. Several points were raised. 

  1. Is it possible to provide more detailed JEV epidemiology in India (number of cases)?

Response: Authors added more information about JEV epidemiology in the manuscript as suggested by the reviewer.

  1. Please mention licensed vaccines in India.

Response: Several Asian countries, including India, have licensed and used a live-attenuated SA 14-14-2 JE vaccine (LAJEV) and JENVAC. Authors added the information of JEV vaccine in the manuscript.

  1. Is JEV vaccines for veterinary use available in India? If yes, please mention its coverage or something like that.

Response: Currently, there is no JEV vaccine available in India for use on animals. ICAR-Indian Veterinary Research Institute, Izatnagar has developed an inactivated JEV vaccine, but its market launch will take some time.

  1. Materials and Table 1: Which tissues of pigs did authors obtain? Can authors clarify?

Response: Tissues of pulled organs like Spleen, Liver, Kidney and Heart.

  1. Table 1: How did authors determine the genotype? Did authors check sequence of E genes of all positive samples? Please clarify.

Response: Yes, based on the E gene of the JEV in PCR using Reference Primer used Forward primer: CCGATCGTCTCCGTTGCGAGCC and Reverse primer: TTTGGCCTTCTTAGCCACAG (http://dx.doi.org/10.4167/jbv.2016.46.4.231).

  1. Fig. 5: The figure legend is missing.

Response: Corrected the Fig. 5 legend.

  1. Fig. 5: If authors can provide isolated year and its source (mosquito, human…) for each data in the phylogenetic tree, that would be quite useful.

Response: A supplementary file has added for this details.

  1. Fig. 6: Were these substitutions synonymous or nonsynonymous?

Response: The substitutions were unique to the newly identified strain of JEV from pig and mosquitos separately.

  1. Fig. 6: If authors can provide the viral proteins containing these substitutions in the figure , it should be helpful.

Response: The data has added in supplementary table 5 and 6

  1. Table 2: If I understand correctly, conventional PCR was conducted only for real-time PCR positive sample. However, table 2 contained the sample for real-time negative and conventional PCR positive.

Response: We appreciate your notice and have corrected it. These samples tested positive both in RT-PCR and conventional PCR. It’s a typographic error.

  1. I cannot find supplementary data.

Response: The supplementary files are attached.

  1. Discussion: Please provide discuss on the genotypic epidemiology in India. Several papers describing that genotype III and I are found, but genotype III is recognized as dominant, can be found.

Response: Authors added more information about JEV epidemiology in the manuscript as suggested by the reviewer.

Reviewer 4 Report

Major comments:

Line 226~228, 159: The authors emphasized that JEV full genome sequencing from pig and mosquito vector belongs to genotype III circulating in districts of Assam, during 2018-2022 , which is same with the virus circulating in the human, but the human isolate 057434 (EF623988) showed 98% nucleotide homology was isolated in India, 2005. This sequence similarity between pig and mosquito JEV isolates, 2018-2022, and human JEV isolate, 2005, might be the continuous genome evolution of JEV in India.

Minor comments:

1.     Line 161:  “---(Fig.3).”  à  “---(Fig.5).”

2.     Line 163:  “ (-----and MZ782743). à The Genebank no. is not correct.

3.     Line 191:  “Mansonia”  à This genus name should be italics

4.     Line 265: The ref 9 is incomplete.

5.     Line 270: The ref 11 is incomplete.

6.     Line 271: The ref 12 is incomplete.

7.     Line 275: The ref 14 is incomplete.

8.     Line 280: The ref 16 is incomplete.

9.     Line 282: The ref 17 is incomplete.

10.  Line 287: The legends of Fig. 2 “ H and “ à?

11.  Line 293: The legends of Fig. 5 “C.”  à?

12.  Line 299: The genus names of Mansonia, Armegeres, Anopheles should be italics.

13.  Line: 18, 229, 295: in Abstract, Conclusion sections, and Fig. 6: The complete genome analysis of two JEV isolates from fig and Culex mosquito were revealed 7 and 20 numbers of points polymorphism of nucleotide alignments of other sequences, respectively. If JEV circulation in the Kamrup and Jorhat and in the same time periods, the JEV was transmitted between pigs and mosquitoes, why these two isolates showed different point polymorphism with each other’s?

Author Response

Response to Reviewers comments

Reviewer 4:

The authors would like to express their sincere thanks to the Editor and the anonymous reviewers for their great comments and suggestions. We are also sincerely thankful to Editor for providing an opportunity to revise and resubmit this manuscript for publication in Vaccines.

Major comments:

Line 226~228, 159: The authors emphasized that JEV full genome sequencing from pig and mosquito vector belongs to genotype III circulating in districts of Assam, during 2018-2022, which is same with the virus circulating in the human, but the human isolate 057434 (EF623988) showed 98% nucleotide homology was isolated in India, 2005. This sequence similarity between pig and mosquito JEV isolates, 2018-2022, and human JEV isolate, 2005, might be the continuous genome evolution of JEV in India.

Response: Thank you for your valuable comments

Minor comments:

  1. Line 161:  “---(Fig.3).”  à  “---(Fig.5).”:

Response: Corrections made in the manuscript as desired by the reviewer

  1. Line 163:  “ (-----and MZ782743). à The Genebank no. is not correct.

Response: Corrections made in the manuscript as desired by the reviewer. Correction done as MZ702743. My sincere thanks to the reviewer once again.

  1. Line 191:  “Mansonia”  à This genus name should be italics.

Response: Corrections made in the manuscript as desired by the reviewer

  1. Line 265: The ref 9 is incomplete.

Response: Corrections made in the manuscript as desired by the reviewer

  1. Line 270: The ref 11 is incomplete.

Response: Corrections made in the manuscript as desired by the reviewer

  1. Line 271: The ref 12 is incomplete.

Response: Corrections made in the manuscript as desired by the reviewer

  1. Line 275: The ref 14 is incomplete.

Response: Corrections made in the manuscript as desired by the reviewer

  1. Line 280: The ref 16 is incomplete.

Response: Corrections made in the manuscript as desired by the reviewer

  1. Line 282: The ref 17 is incomplete.

Response: Corrections made in the manuscript as desired by the reviewer

  1. Line 287: The legends of Fig. 2 “ H and “ à?

Response: Corrections made in the manuscript as desired by the reviewer. H and E represents hematoxylin and eosin.

  1. Line 293: The legends of Fig. 5 “C.”  à?

Response: Corrections made in the manuscript as desired by the reviewer

  1. Line 299: The genus names of Mansonia, Armegeres, Anopheles should be italics.

Response: Corrections made in the manuscript as desired by the reviewer

  1. Line: 18, 229, 295: in Abstract, Conclusion sections, and Fig. 6: The complete genome analysis of two JEV isolates from pig and Culex mosquito were revealed 7 and 20 numbers of points polymorphism of nucleotide alignments of other sequences, respectively. If JEV circulation in the Kamrup and Jorhat and in the same time periods, the JEV was transmitted between pigs and mosquitoes, why these two isolates showed different point polymorphism with each other’s?

Response: Area wise and sample wise, there must be variations in JEV strains. In our study, we pointed out that genotype-III from the same geographical area and different sources may have different point polymorphisms.

Round 2

Reviewer 2 Report

·         The most important problem with the current manuscript is authors cannot claim to have sequenced the complete Genome without confirming the terminal sequences by RACE. Authors must change the title to reflect what was actually done. Authors should consider the fact that NGS may not capture the NTRs of both ends of the genome, which is required for complete genome sequencing.

·            What was the time/period of mosquito collection

·         Paragraph 2.2, authors need to replace 'isolated' since virus isolation is a completely different thing that does not include RNA extraction

·         Authors need to include the Ct value cut off in the manuscript

Author Response

Response to Reviewer’s comments:

  • The most important problem with the current manuscript is authors cannot claim to have sequenced the complete Genome without confirming the terminal sequences by RACE. Authors must change the title to reflect what was actually done. Authors should consider the fact that NGS may not capture the NTRs of both ends of the genome, which is required for complete genome sequencing.

Response: The critical observation by the esteemed reviewer is absolutely correct. We understand that RACE is the high-throughput technology to complete the end sequence of RNA transcripts. However, current technological development in Illumina sequencing tools has the property to sequence the N terminal of the RNA virus during library preparation. However, on recommendation of the esteemed reviewer we changed the title of the manuscript as " Japanese Encephalitis Virus genotype III strains detection and genome sequencing from Indian pig and mosquito vector".

                  Also, removed the “complete/ full/ whole genome” word from the title as well as manuscript where it represents our JEV sequences.

  • What was the time/period of mosquito collection

Response: The mosquito samples were collected from Kamrup and Jorhat districts in accordance with the areas of pig farms during the dusk period of the day. Sampling of mosquito took place from 2018 to 2020, as did sampling for other pig blood, tissue and cerebral fluid.

  • Paragraph 2.2, authors need to replace 'isolated' since virus isolation is a completely different thing that does not include RNA extraction

Response: Corrections made in the manuscript as desired by the reviewer

  • Authors need to include the Ct value cut off in the manuscript

Response: Corrections made in the manuscript as desired by the reviewer

Reviewer 3 Report

Authors revised the manuscript accordingly.

Author Response

Response: That’s great to hear! We are so happy and thank you so much for your kind words and appreciation.

Round 3

Reviewer 2 Report

Authors have addressed most of my concerns.

I suggest authors indicate clearly the time of mosquito collection (For example between 12 - 2 am) 

Author Response

Authors have addressed most of my concerns.

: That’s great to hear! We are so happy and thank you so much for your kind words and appreciation.

I suggest authors indicate clearly the time of mosquito collection (For example between 12 - 2 am) 

Response: The time period of mosquito collection was between 5.00 PM to 7.00 PM. Which was also mention in the manuscript as desired by the reviewer.

I would like to express my gratitude to the reviewer for making the manuscript perfect.